# Machine learning code snippets semantic classification

Valeriy Berezovskiy[1], Anastasia Gorodilova[1], Ekaterina Trofimova[1] and Andrey Ustyuzhanin[2,3]

[1] HSE University, Moscow, Russia
[2] Institute for Functional Intelligent Materials, National University of Singapore, Singapore
[3] Constructor University, Bremen, Germany

## ABSTRACT

Program code has recently become a valuable active data source for training various data science models, from code classification to controlled code synthesis. Annotating code snippets play an essential role in such tasks. This article presents a novel approach that leverages CodeBERT, a powerful transformer-based model, to classify code snippets extracted from Code4ML automatically. Code4ML is a comprehensive machine learning code *corpus* compiled from Kaggle, a renowned data science competition platform. The *corpus* includes code snippets and information about the respective kernels and competitions, but it is limited in the quality of the tagged data, which is ~0.2%. Our method addresses the lack of labeled snippets for supervised model training by exploiting the internal ambiguity in particular labeled snippets where multiple class labels are combined. Using a specially designed algorithm, we effectively separate these ambiguous fragments, thereby expanding the pool of training data. This data augmentation approach greatly increases the amount of labeled data and improves the overall quality of the trained models. The experimental results demonstrate the prowess of the proposed code classifier, achieving an impressive F1 test score of ~89%. This achievement not only enhances the practicality of CodeBERT for classifying code snippets but also highlights the importance of enriching large-scale annotated machine learning code datasets such as Code4ML. With a significant increase in accurately annotated code snippets, Code4ML is becoming an even more valuable resource for learning and improving various data processing models.

## INTRODUCTION

The demand for machine learning (ML) code as a source of data has recently grown among data scientists (*Agashe, Iyer & Zettlemoyer, 2019*; *Quaranta, Calefato & Lanubile, 2021*). However, training a deep learning model often requires large-scale annotated *corpora*. Thus, a well-annotated ML-specific code data set can facilitate the expansion of data processing methods.

Code4ML, a collection of ≈2.5 million snippets of machine learning code introduced by *Drozdova et al. (2022)*, is limited in a case of annotated data—only 0.2% of the collected snippets are manually labeled. The authors apply the sequence of support vector machines (SVM) models for snippets automatic classification. First, an SVM with a linear kernel



Corresponding author
Ekaterina Trofimova,
ktrofimova@protonmail.com

model is trained on the Code4ML labeled data, then the predictions on an unlabeled part are used for a cross-validation training of the RBF-based SVM model. However, given the size of the data *corpus*, the automatic classification of Code4ML snippets needs more accurate verification.

Our work aims to enrich the Code4ML *corpus* with high-quality, accurate annotation. We explore various machine learning models for code snippet classification to achieve this. We begin with classical methods based on TF-IDF code transformation, such as linear models and ensembles. Subsequently, we delve into modern transformer models, initially introduce by *Vaswani et al. (2023)*, significantly enhancing classification performance compared to baseline approaches. Among the transformer models, our best-performing classifier is a fine-tuned CodeBERT model, as introduced by *Feng et al. (2020)*. We also present a novel data augmentation algorithm based on separating code chunks containing snippets of different semantic types. This augmentation strategy effectively expands the training set and elevates the quality of the final model.

The rest of the article is organized as follows. Section 'Literature Review' comprehensively reviews the existing literature on code classification. The code snippets *corpus* with several pre-processing tools to prepare source codes for the classification and data structure hypothesis are discussed in 'Data and Metrics'. The following section describes the dataset employed for training and evaluation. In 'Experiments and Results', we present the results of fine-tuning CodeBERT and compare them against the baseline models. The 'Discussions' section involves analyzing the obtained results. Finally, the section 'Conclusion' highlights our contributions and avenues for future research.

## LITERATURE REVIEW

Many approaches to code analysis rely heavily on abstract syntax tree (AST) representation of source code (*Wei & Li, 2017*; *Zhang et al., 2019*; *Azcona et al., 2019*). ASTs are abstract and exclude characters such as punctuation. At the same time, combining such a representation with neural networks allows linguistic and syntactic information collection. The mentioned research for learning the code's semantic features is not intended to study the semantic features of different programming languages. *Wang et al. (2022)* propose an application of a self-attention mechanism and a graph convolutional neural network (GCN) for extracting global and local features of the AST sequence, respectively. This approach eliminates the differences between programming languages in the problems of classifying interlanguage programs. *Puri et al. (2021)* show the competitiveness of GCN, as well as of C-BERT (*Buratti et al., 2020*), a model pre-trained with C programs and fine-tuned on AST, in solving the problem of code classification. C-BERT leverages the bidirectional transformer (BERT) model (*Devlin et al., 2018*) to effectively extract AST features from the source code. *Yang, Jin & Dou (2023)* extends the graph neural network paradigm to handle heterogeneous and directed edges in the code's abstract syntax tree.

AST implementation for programming code embeddings creation works excellent for the algorithmic datasets, such as POJ-104 (*Mou et al., 2014*), CodeNet (*Puri et al., 2021*), because such code has a lot of nested loops and other complex constructs. Therefore, the

AST representation of such code is very informative. Meanwhile, the ML code is often represented *via* pipelines, sequences of repeating typical patterns, such as code snippets for loading data, training a model, or evaluating metrics. Thus, we argue that AST representation for ML-related datasets is uninformative by calculating the average depth of the Code4ML syntactic snippet tree, which is almost twice less than in the POJ-104 dataset.

# DATA AND METRICS

## Dataset

We use Code4ML as a *corpus* of ML code snippets, including ≈8 thousand human-curated annotated unique snippets. For snippets annotation, the authors introduce Taxonomy Tree, a set of 11 high-level categories (*Data Transform, Exploratory Data Analysis, Data Extraction, Environment, Data Export, Debug, Visualization, Hyperparam Tuning, Model Train, Model Interpretation, Model Evaluation*) and ≈7 semantic types corresponding to each level (for example, Save Model subclass of Model Train class). There is also a separate category *Other*, reflecting snippets whose semantic types can not be identified within the Taxonomy Tree.

The semantic type of the snippet in Code4ML is the subtask that the snippet solves. Figures 1 and 2 show examples of a snippet of different types.

Each marked-up snippet is provided with a *too_long* flag and a *mark*. *too_long* indicates whether the snippet represents unambiguous code, that is, can not be divided into several subsnippets of different semantic types. The approximate proportion of code blocks with *too_long* = '*yes*' to ones with '*no*' is 6:1. The majority of the snippets have the highest *mark*, a confidence level of a semantic snippet type provided by a human assessor.

**Data pre-processing**. For all experiments, we use a set of pre-processed snippets with the maximum confidence level for the experiments. The snippets of category *Other* are also removed. The resulting dataset has 5,152 unique snippets. (We will call these snippets raw.) The data is split into two parts in all the experiments: 60% for train and validation and 40% for results testing.

Code4ML includes only the code written in Python due to the popularity of this programming language in the data science community. The following code transformations have been used to eliminate redundant information from the snippet and make it easier for ML models to work with it:

1) Comments deleting
2) Module imports removing
3) Empty strings removing
4) The remaining tokens are separated by a space
5) The programmer symbols are separated by spaces

Importing various modules and libraries or commenting on the code does not change the semantic type of the code snippet (except only imports in the code cell). Therefore, removing them helps the model correctly identify the semantic snippet type. Both transformations are done using regular expressions: Python's module imports and

```
#file to save
filename = 'finalized_model.sav'
#model saving
pickle.dump(model, open(filename, 'wb'))
```

**Figure 1** **Code4ML snippet of** *Model Train* **class** *Save Model* **semantic type example.**

```
#adding column `month`
df['month'] = [x.month for x in df['Open_date']
#adding column `year`
df['year'] = [x.year for x in df['Open_date']
```

**Figure 2** **Code4ML snippet of** *Data Transform* **class** *Feature Engineering* **semantic type example.**

commenting have a simple syntax that is easily recognized by the *re* module. Figures 3 and 4 illustrate the difference between processed and raw code blocks.

## Metrics

To assess the quality of the semantic code snippets classification models, we apply weighted precision, recall (Eq. (1)) and F1-score (Eq. (2)). We choose precision and recall as the most interpretable metrics and F1 as a validation suitable for unbalanced data.

$$\text{precision} = \frac{\text{TP}}{\text{TP} + \text{FP}}, \ \text{recall} = \frac{\text{TP}}{\text{TP} + \text{FN}},$$ (1)

where TP—true positive, TN—true negative, FP—false positive, FN—false negative rates.

$$F_\beta = (1 + \beta^2) \frac{\text{precision} \times \text{recall}}{\beta^2 \text{precision} + \text{recall}}.$$ (2)

## SEMANTIC CLASSIFICATION MODELS

To classify the ML snippets into semantic types, we leverage the power of the transformer models without focusing on the AST representation of the source code. BERT-family models use the input information to produce a feature representation set for downstream tasks. In this section, we briefly explain the mechanism of CodeBERT by *Feng et al. (2020)*, the pioneer pre-trained model for both natural language and programming language input (see Fig. 5). We also discuss an augmentation algorithm that divides *too_long* flagged code snippets into subsnippets.

## BERT and CodeBERT models

Large pre-trained transformer-based models, such as BERT, can be called the current standard in various natural language processing (NLP) tasks. BERT is a self-supervised model whose workflow consists of pre-training and fine-tuning. The first stage solves masked language modeling (MLM) and the following sentence prediction tasks while fine-tuning is used for downstream applications.

CodeBERT is pre-trained on CodeSearchNet dataset (*Husain et al., 2019*) with different objectives (MLM or RTD) and settings: from scratch and with the initialization of

```
#get character data columns
notNumTypeCol = [col for col in
                 train_df.columns if
                 train_df[col].dtype ==
                 dtype('O')]
```

**Figure 3  Raw code snippet.**               

```
['notNumTypeCol', '=', '[', 'col',
'for', 'col', 'in', 'train', '_',  'df',
'.', 'columns', 'if', 'train', '_',
'df', '.', 'dtype', '==', 'dtype', '(',
''O'', ')', ']']
```

**Figure 4  Preprocessed code snippet.**     

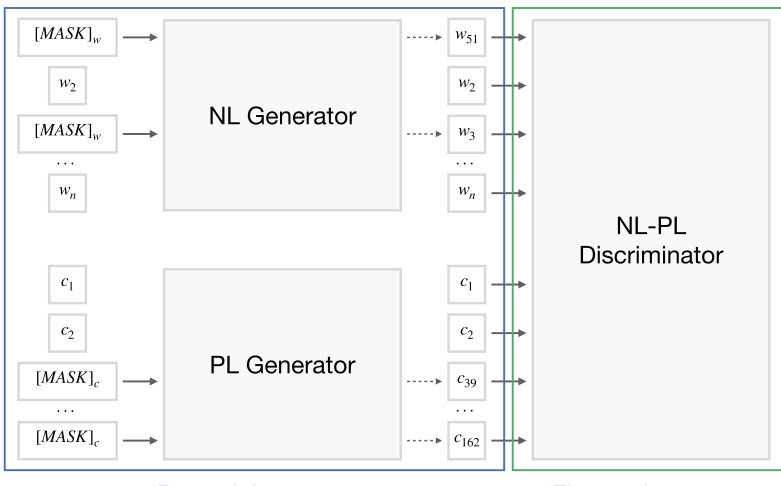

**Figure 5  CodeBERT architecture scheme about the replaced token detection objective.** Based on original article by *Feng et al. (2020).*           

RoBERTa (*Liu et al., 2019*) parameters. To make the model able to classify ML code into semantic types, we fine-tune CodeBERT on Code4ML labeled data. We use multiple objectives with the RoBERTa weights initialization configuration of the model as it shows the best results on the Python data according to the original article. We also compare the fine-tuned CodeBERT with pre-trained BERT, fine-tuned on the ML snippets[1]. The experiments and results are described in 'Experiments and Results'.

## Dataset augmentation

As shown earlier, a significant portion, approximately 20%, of the data is not amenable to unambiguous interpretation as valid code. Consequently, such code snippets cannot be effectively employed for model validation, as their class membership cannot be unequivocally determined. As a result, predictions made by the model in these instances cannot be definitively labeled as correct or incorrect. Notably, including such ambiguous code in the training set only worsens the model's performance on new data, resulting in a

[1] For the experiments, we use models and tokenizers from the transformers library https://huggingface.com. The code can be accessed *via* https://zenodo.org/record/8250804.

decrease in overall quality. However, it has been observed that some data instances contain code segments from several classes, often representing a sequence of code fragments, each representing a fragment of a particular semantic type (see Fig. 6).

To deal with such multi-class snippets, we propose a partitioning algorithm (Algorithm 1) that intelligently splits a single snippet into several subsnippets. It starts with finding all possible ways of breaking one snippet into sequences of subsnippets.

The algorithm seeks to determine the optimal partition that minimizes the model's confidence for each subsnippet while considering the average model confidence across all subsnippets in the division. Here, confidence refers to the probability of a subsnippet belonging to a specific semantic type, calculated using the softmax operation.

The correctness of the split is checked separately: it is unacceptable to split the snippet in the place where a line break occurred due to a multi-line function call, or a line break; this usually happens in code to comply with PEP-8. Such splits, influenced by code formatting rather than semantic boundaries, are deemed unacceptable to maintain the integrity and accuracy of the generated subsnippets.

By employing this partition algorithm, we can effectively handle complex multi-class snippets and generate meaningful subsnippets. These are valuable training data to enhance model performance on ambiguous code instances. The resulting partitioned subsnippets not only improve the quality of the trained model but also facilitate more accurate code classification and synthesis tasks.

Figure 7 is an example of a snippet that contains code from two classes: *read_csv* and *prepare_x_and_y*. After using the snippet partition algorithm, it will be divided into two snippets, which are shown in Figs. 8 and 9.

The initial model's confidence in the entire snippet is relatively low due to its multi-class nature, making accurate classification challenging. While an alternative option for splitting the snippet into a snippet comprising the first two lines and another with the third line exists, this approach would yield suboptimal model confidence in one of the snippets. We obtain two high-quality snippets from the initially ambiguous code through the partition algorithm, which serves as valuable training data to enhance the model's performance.

Regarding computational efficiency, we acknowledge that the running time of our algorithm can be exponential due to the large number of possible partitions. To mitigate this, we introduce the **max_lines** parameter to the algorithm, controlling the maximum number of lines in a code snippet. If the snippet exceeds the threshold **max_lines** = 20, the algorithm retains the original form of the snippet without searching for an optimal partition. Consequently, the algorithm operates within a few seconds for code snippets with up to approximately 20 lines, making it highly efficient for most cells in Jupyter notebooks, which often contain concise code snippets.

## EXPERIMENTS AND RESULTS

We compare the transformer-based and classical ML models to classify the ML code snippets. In the case of classical ML models, the hyper-parameters choice is made using the three-folds cross-validation method, aiming to maximize the model's accuracy. We use

```python
# Taxonomy type: read_csv
df = pd.read_csv('data.csv')
# Taxonomy type: prepare_x_and_y
y = df['target']
X = df.drop(['target'])
# Taxonomy type: split
X_tr, X_t, y_tr, y_t = train_test_split(X, y)
# Taxonomy type: train_model
model = svm.SVC().fit(X_tr, y_tr)
# Taxonomy type: predict_on_test
y_pred = model.predict(X_t)
# Taxonomy type: compute_test_metric
acc_test = accuracy_score(y_t, y_pred)
```

**Figure 6** **An example of a snippet that contains code from different classes (classes are specified in the comments).**

---

**Algorithm 1 Snippet partition algorithm.**

**Require:** $x_i$

**Ensure:** $\{x_{i,1}, x_{i,2}, \ldots\}$          ▷ Snippet

    all_partitions ← all possible partitions      ▷ Snippet partition

    partition_probs ← ∅

  **for** partition ∈ all_partitions **do**

    **if** partition is correct **then**

      min_prob ← minimal subsnippet confidence

      mean_prob ← average supsnippet confidence

      partition_probs ← partition_probs ∪ (min_prob, mean_prob)

    **end if**

  **end for**

**return** $\arg\max_{\text{partition}}$ (partition_probs)

---

```python
df = pd.read_csv('data.csv')
y = df['target']
X = df.drop(['target'])
```

**Figure 7** **Code4ML snippet before partition.**

```python
df = pd.read_csv('data.csv')
```

**Figure 8** **The first resulted subsnippet.**

TF-IDF embedding to extract the numerical features from the pre-processed code blocks as input to the classical models. In all experiments, we use a fixed random seed for the reproducibility of the results.

```
y = df['target']
X = df.drop(['target'])
```

**Figure 9** The second resulted subsnippet.

**C-support vector classification**. For the c-support vector classification (SVC) model (*Platt, 1999*), the following hyper-parameters are chosen with the gridsearch technique: *C*, indicating the strength of the model's regularization, *kernel*, the separating hyperplane type, and *gamma*, kernel coefficient. The best results are gained with $C = 1.668$, selected from 10 logarithmically spaced points from 0.01 to 100, linear kernel out of linear, polynomial, and RBF types, and *gamma* = 0.01, chosen from the same logarithmic vector as *C* hyper-parameter.

**Logistic regression**. The best maximum-entropy classification method (*Bishop, 2006*) results go with $C = 100.0$ from the same logarithmic vector and 12 norm of the penalty out of binary choice: 12 or none penalty.

**Random forest**. For random forest ensemble model (*Breiman, 2001*), the crucial hyper-parameters are *n_estimators*, corresponding to the number of the trees in the ensemble, *max_depth*, the maximum depth of the tree, *min_samples_split*, indicating the minimum required several objects to split a node and *min_samples_leaf*, the minimum required a number of objects at a leaf node. The best random forest performance is achieved with *n_estimators* = 100 out of [100, 200, 400, 800], *max_depth* = *None*, which means that nodes are expanded until all leaves are pure or until all leaves contain less than *min_samples_split* samples, outperforming configurations with other grid values (*max_depth* = 5 an *max_depth* = 20), *min_samples_split* = 8, *min_samples_leaf* = 1 out of [2, 8, 32], [1, 4, 16], correspondingly.

**Gradient boosting**. For gradient boosting ensemble model (*Friedman, 2001*), the search is done on the [20, 40] grid for *n_estimators*, [2, 4, None] for *max_depth*, [0.01,0.1,1] for *learning_rate* (lr), the gradient step size. The model with *n_estimators* = 40, *max_depth* = 4, *learning_rate* = 0.1 outperforms its variations with different choices of hyper-parameters.

The hyper-parameters choice is made with stratified three-folds cross-validation for the above mentioned experiments. Thus, the distribution of classes in each split is preserved, and each group is kept within a single partition. Table 1 summarizes the test results of these models. SVC gives the best result among the classical models based on F-1 score. Ensemble methods struggle to extract much additional information from the features obtained by TF-IDF transformation because it is not informative enough in the case of working with program code.

**The pre-processing impact on ML models performance.** Table 1 also illustrates the influence of data pre-processing on ML models' performance. We train and validate the models on raw code snippets as well. The greeds for hyper-parameters remain the same. The best results are reached with $C = 4.641$, *gamma* = 0.01 and linear kernel for SVC, $C = 100.0$ and *l2* penalty for Logistic regression, *n_estimators* = 400, *max_depth* = *None*,

**Table 1  ML models performance on raw test data.** The best results are highlighted in bold.

| | Processed data | | | Raw data | | |
|---|---|---|---|---|---|---|
| Model | F1-score | Precision | Recall | F1-score | Precision | Recall |
| Logistic regression | 0.835 | 0.853 | 0.828 | 0.488 | 0.640 | 0.481 |
| Random forest | 0.833 | **0.864** | 0.819 | 0.453 | 0.650 | 0.436 |
| Gradient boosting | 0.751 | 0.772 | 0.755 | 0.350 | 0.550 | 0.369 |
| SVC | **0.839** | 0.856 | **0.832** | 0.458 | 0.605 | 0.456 |

$min\_samples\_split = 2$, and $min\_samples\_leaf = 1$ for Random forest and $n\_estimators = 40$, $max\_depth = None$, $learning\_rate = 0.1$ for boosting. One can see that pre-processing leads to a significant quality increase.

## BERT and CodeBERT

As mentioned earlier, we fine-tune both BERT and CodeBERT models on the Code4ML dataset labeled part with a confidence equal to 5. The train and validation parts are 40% and 20%, respectively. Both models require no additional pre-processing of the code data. We utilize the standard tokenizers with parameters $add\_special\_tokens = True$, $padding = max\_length$, $truncation = True$, and $max\_length = 512$ to handle input sequences of up to 512 tokens.

For the training process, we employ a batch size of 32, AdamW optimizer, and conduct training for 50 epochs. We adopt a linear scheduler with a warmup, a common approach for transformer models. During the initial ten epochs of training, the learning rate gradually has increased from zero to 5e−5, and in the subsequent 40 epochs, it linearly has decreased back to zero. The scheduler step is applied to each batch of the training set. The learning rate value 5e−5 is selected considering both models had already been pre-trained on extensive natural and programming language datasets.

To adapt both neural networks for code classification, we adjust the number of vertices in the last classification layer and unfroze all weights for training. The performance curves of the CodeBERT model can be found in "Snippet Splitting Algorithm Performance Assessment", while "Snippets Classification Algorithm Confidence Assessment" justifies the prediction confidence of the CodeBERT.

The test scores of the experiments are shown in Table 2. One can notice that transformer-based models fine-tuned on the code snippets demonstrate a quality increase compared to the SVM model. Due to pre-training on the program code, including the Python code, CodeBERT outperforms its predecessor model. The overall F1-score of the CodeBERT model is ≈87%.

**Application of the augmentation algorithm.** As mentioned, the results are obtained on ≈5K processed (in case of ML models) or raw (in case of BERT and CodeBERT) high-marked code snippets. We explore the capability of the snippet partition algorithm by analyzing the quality of CodeBERT fine-tuned on the different combinations of code

**Table 2 Test performance of the transformer-based models fine-tuned on the code snippets with** *mark* = 5**. The best results are highlighted in bold.**

| Model | F1-score | Precision | Recall |
|-------|----------|-----------|--------|
| BERT | 0.852 | 0.854 | 0.858 |
| CodeBERT | **0.871** | **0.874** | **0.873** |

blocks with the highest *mark* and the ones processed by the Algorithm 1 with the *mark* = 4.

When dealing with ambiguously interpreted code snippets, it is essential to be careful when using them. As mentioned earlier, such snippets are only suitable for inclusion in the training set since their ambiguous nature makes them unsuitable for the test set. Similarly, the snippets generated by the augmentation algorithm should be subjected to human verification before inclusion in the test set. Since we can not guarantee the model's predictions on these augmented snippets, they are exclusively added to the training set to prevent compromising the validity of the test results. The validation and test sets remain consistent with previous experiments, ensuring a fair comparison of model performance.

To further elaborate, we select 1,414 raw snippets with a confidence mark of four, and through the application of the augmentation algorithm, 2,152 augmented snippets are generated from this initial set. These additional snippets are purely for enhancing the training data, increasing the model's ability to handle ambiguous and varied code snippets efficiently. Careful handling of the augmented snippets ensures the robustness and accuracy of the trained model on new and unseen data.

CodeBERT shows the best results of $\approx 89\%$ F1 score on the mix of the augmented snippets with *mark* = 4 and snippets of higher confidence. Such data configuration implies $\approx 40\%$ training dataset augmentation. The details on CodeBERT performance comparison are provided in "Snippet Splitting Algorithm Performance Assessment".

## DISCUSSION

The results of our experiments reveal promising advancements in machine learning code snippets annotation, mainly through the application of transformer-based models like BERT and CodeBERT.

Firstly, the fine-tuning of transformer-based models on the Code4ML dataset significantly enhances the classification quality compared to traditional approaches like SVM. With their ability to capture complex contextual information, the transformer models demonstrate a superior understanding of code semantics, leading to more accurate predictions.

Moreover, the integration of CodeBERT, a transformer model pre-trained on programming and natural language data, yields remarkable improvements over BERT. Leveraging the vast amount of programming language-specific context, CodeBERT excels in code-related tasks, outperforming its predecessor model, BERT.

The proposed extension algorithm improves the model's performance on ambiguously interpreted code fragments. We create high-quality subsnippets that contribute to an

enriched training set by intelligently splitting snippets from multiple classes. This extension strategy effectively solves the problem of limited annotated data, resulting in improved model performance and code classification accuracy.

Compared to the results reported in *Drozdova et al. (2022)*, our baseline model (SVM) achieves a remarkable improvement of 14 percentage points in the F1 score. The previous study obtained an F1 score of 68.9%, and although they achieved further enhancements using pseudo-labels, our superior data preprocessing techniques significantly improve performance without relying on pseudo-labels.

Moreover, incorporating CodeBERT and developing our augmentation algorithm contribute to an additional increase in the F1 score. As a result, our models demonstrate substantially better performance without resorting to pseudo-labels from the remaining dataset. This enhancement enables our models to make more objective predictions on unlabeled data, as it has not been artificially included in the training set.

However, we acknowledge the limitation of our algorithm's exponential running time for snippets with many lines. Optimizing the partition algorithm to handle larger snippets efficiently would be beneficial as a future direction.

Furthermore, we emphasize the importance of cautiously handling ambiguous and augmented snippets. It is essential to restrict their inclusion only to the training set, as using them in the test sample may compromise the integrity of the test results.

## CONCLUSION

This study highlights the promising potential of modern natural language processing models in addressing the challenging problem of machine learning code classification. Through comprehensive comparisons between classic ML models and fine-tuned transformer-based models, we have demonstrated the superiority of CodeBERT, which stands out due to its multi-modal structure, enabling effective processing of programming code and benefiting from pre-training on Python code.

The proposed method significantly enhances the automatic classification of machine learning code snippets, leading to notable improvements in performance. Consequently, the large-scale Code4ML *corpus* is enriched with high-quality annotations (*Drozdova et al., 2022*), serving as a valuable resource for various machine learning challenges, including code generation, auto-completion, and other ML tasks.

Our findings reinforce the significance of effectively leveraging transformer-based models and data augmentation techniques to tackle code classification challenges. By intelligently handling ambiguously interpreted code snippets and efficiently expanding the training data, we advance the accuracy and robustness of code classification models.

The done research paves the way for further investigations into adapting transformer-based models for other programming languages and exploring novel approaches to optimize the partition algorithm for larger snippets.

In conclusion, this study contributes to advancing automated code understanding and generation, opening up exciting opportunities for future research in machine learning code

| Table A1 CodeBERT performance on test data. The best results are highlighted in bold. | | | |
|---|---|---|---|
| Data | F1-score | Precision | Recall |
| Raw snippets with *mark* = 5 (rs5) | 0.871 | 0.874 | 0.873 |
| Raw snippets with *mark* = 4 (rs4) | 0.593 | 0.591 | 0.638 |
| Augmented snippets with *mark* = 4 (as4) | 0.709 | 0.705 | 0.748 |
| rs5 and rs4 | 0.872 | 0.874 | 0.875 |
| rs5 and as4 | **0.888** | **0.890** | **0.891** |

classification. The enriched Code4ML *corpus* and the state-of-the-art performance achieved by CodeBERT underscore the potential of modern NLP models in empowering code analysis tools.

## SNIPPET SPLITTING ALGORITHM PERFORMANCE ASSESSMENT

In this appendix, we evaluate the impact of the snippet splitting algorithm presented in "Dataset augmentation" on the performance of CodeBERT.

We assess the Algorithm 1 by running the following tests:

- Training and Validation CodeBERT model on the raw snippets with *mark* = 5
- Training and Validation CodeBERT model on the raw snippets with *mark* = 4
- Training and Validation CodeBERT model on the augmented snippets with *mark* = 4
- Training and Validation CodeBERT model on the raw snippets with *mark* = 5 and *mark* = 4
- Training and Validation CodeBERT model on the raw snippets with *mark* = 5 and augmented snippets with *mark* = 4

Table A1 provides the comparison of CodeBERT performance trained and validated on different dataset parts. Figures A1–A4 demonstrate the difference in loss and metrics curves of a CodeBERT model fine-tuned on various data configurations. One can notice the superiority of the model trained on augmented snippets with *mark* = 4 over raw snippets with *mark* = 4, as well as the CodeBERT trained on raw snippets with *mark* = 5 and augmented snippets with *mark* = 4 over the one trained on raw snippets with *mark* = 5 and *mark* = 4. Thus, applying the augmentation algorithm to snippets with a lower mark turns them into higher-quality data for training. By combining the raw snippets with *mark* = 5 and the augmented snippets with *mark* = 4, we achieve the highest classification quality in our experiments. The resulting model demonstrates the best performance and provides its predictions on *Drozdova et al. (2022)*. For every snippet, probabilities are supplied, indicating the likelihood of it belonging to each class.

## SNIPPETS CLASSIFICATION ALGORITHM CONFIDENCE ASSESSMENT

In this appendix we do the snippet probability analysis.

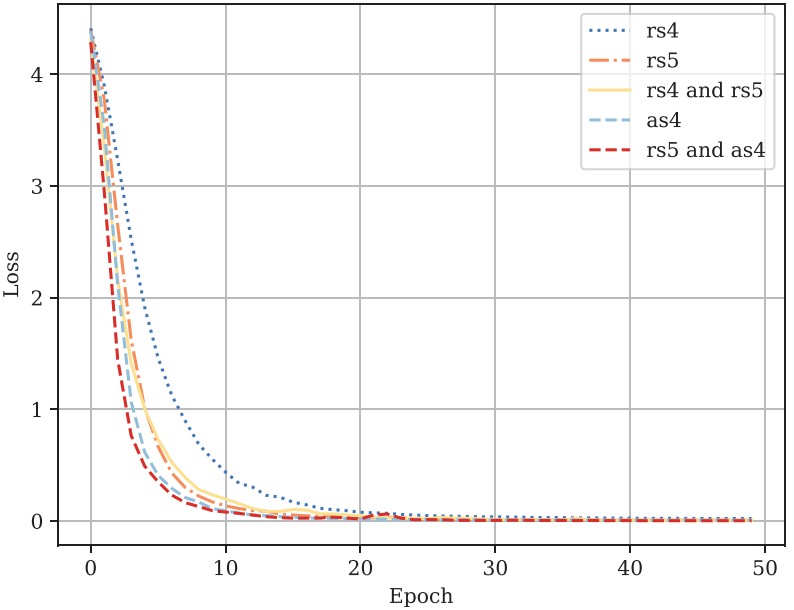

**Figure A1  Fine-tuned CodeBERT training loss curves.**

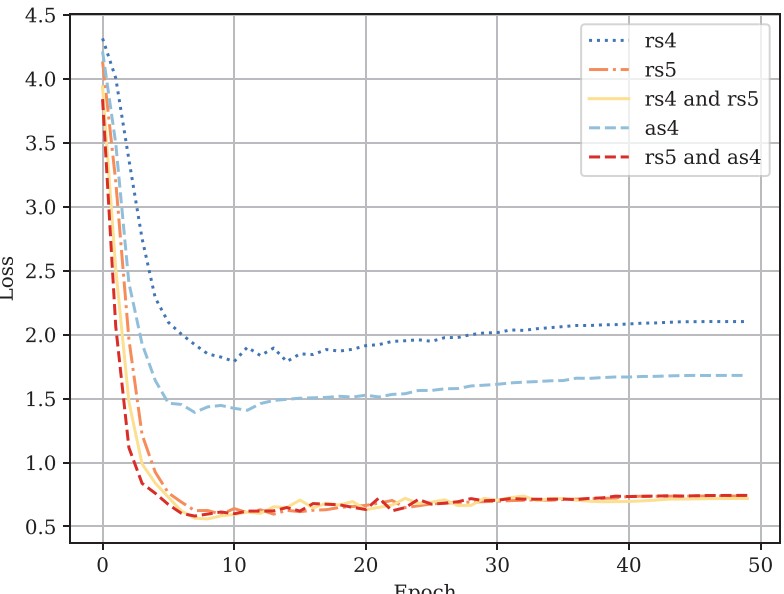

**Figure A2  Fine-tuned CodeBERT validation loss curves.**

The following experiments are carried out on four datasets (train, test, val, and all unmarked snippets). For each value $p$ from 0 to 1 (with step = 1e−5), we calculate the proportion of the samples with the model class probability $\leq p$. Thus, $p$ is the probability threshold. More formally, the value of $f(p)$ is calculated for each $p$ as follows:

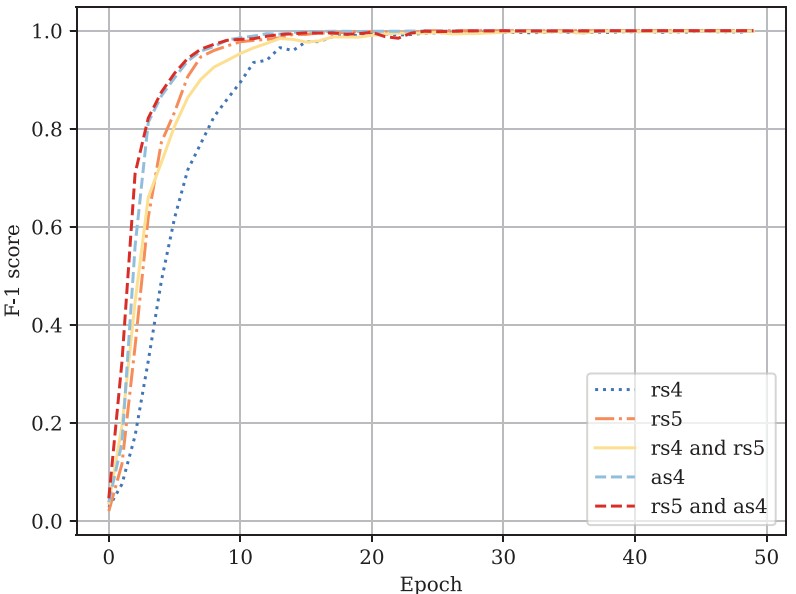

**Figure A3  Fine-tuned CodeBERT training F1-score curves.**

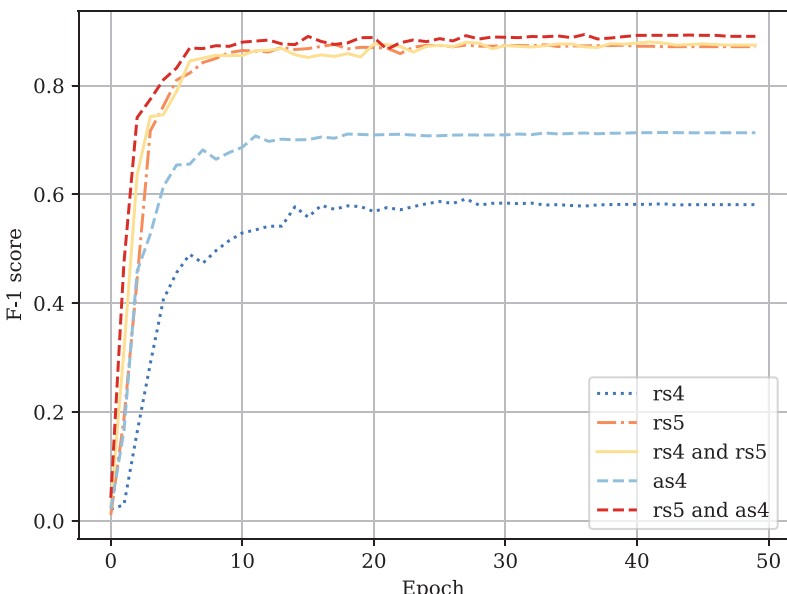

**Figure A4  Fine-tuned CodeBERT validation F1-score curves.**

$$f(p) = \frac{|X_p|}{|X|}$$

$$X_p = \{x \in X \mid P(x) <\,= p\},$$

where $p \in (0, 1)$, $X$ − whole selection, $P(\cdot)$−class probability.

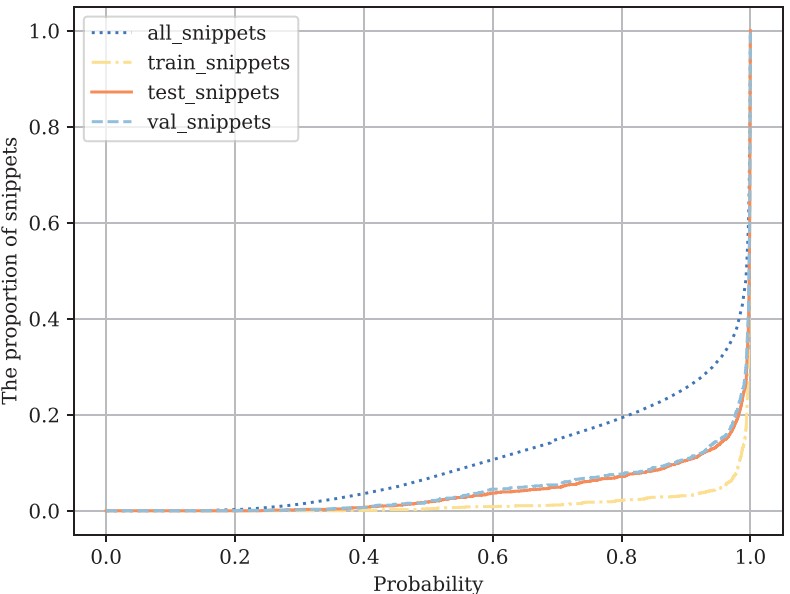

**Figure B1 Comparison of class probabilities on labeled and all other data.**

Figure B1 shows that in the training sample, the model is very confident in its predictions. On the test sample and unlabeled data, the confidence is lower but still relatively high, which shows the generalizing ability of the model, for example, for the snippets where the class probability is ≈0.1. These are those snippets, examples of which were not in the marked-up data: some custom functions that are difficult to attribute to one code class.

### Funding
The publication was supported by the grant for research centers in the field of AI provided by the Analytical Center for the Government of the Russian Federation (ACRF) in accordance with the agreement on the provision of subsidies (identifier of the agreement 000000D730321P5Q0002) and the agreement with HSE University No. 70-2021-00139. The funders had no role in study design, data collection and analysis, decision to publish, or preparation of the manuscript.

### Grant Disclosures
The following grant information was disclosed by the authors:
Analytical Center for the Government of the Russian Federation (ACRF): 000000D730321P5Q0002.
HSE University: 70-2021-00139.

### Competing Interests
The authors declare that they have no competing interests.

## Author Contributions

- Valeriy Berezovskiy conceived and designed the experiments, performed the experiments, analyzed the data, performed the computation work, prepared figures and/or tables, authored or reviewed drafts of the article, and approved the final draft.
- Anastasia Gorodilova conceived and designed the experiments, performed the experiments, analyzed the data, performed the computation work, prepared figures and/or tables, and approved the final draft.
- Ekaterina Trofimova conceived and designed the experiments, analyzed the data, prepared figures and/or tables, authored or reviewed drafts of the article, and approved the final draft.
- Andrey Ustyuzhanin conceived and designed the experiments, authored or reviewed drafts of the article, and approved the final draft.

## Data Availability

The code is available at Zenodo: Valeriy Berezosvkiy. (2023). ketrint/ml-snippets-classification: v1.0.0 (v1.0.0). Zenodo. https://doi.org/10.5281/zenodo.8250804.

The enriched with annotations code snippets dataset is available at Zenodo: Anastasia Drozdova, Polina Guseva, Ekaterina Trofimova, Anna Scherbakova, Andrey Ustyuzhanin, Anastasia Gorodilova, & Valeriy Berezovskiy. (2022). Code4ML: a Large-scale Dataset of annotated Machine Learning Code (1.0.1) [Data set]. Zenodo. https://doi.org/10.5281/zenodo.7733823.

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
