# Peer review of "Machine learning code snippets semantic classification"

_PeerJ Computer Science, doi:10.7717/peerj-cs.1654_

## Round 0.1 · original submission · Major Revisions

In this paper, reviewer 2 has suggest two references. If you feel their inclusion is not necessary to this work, pls don't consider them in the revised version. It will not impact my decision.

Reviewer 1 ·

Basic reporting

- The paper is well written and very clear
- Results are presented with details.

Experimental design

- Extensive experiments are perfomed

Validity of the findings

- This paper has novelty and something that is very new. Such techniques help in building faster datasets by having meaningful embeddings.

Cite this review as

Reviewer 2 ·

Basic reporting

The paper has been written in professional English and explains the problem statement and background in detail. Paper also provides sufficient details on experimental set up and results.

Two relevant literature references are missing:

1) BERT paper:

BERT: Pre-training of Deep Bidirectional Transformers for Language Understanding
Jacob Devlin, Ming-Wei Chang, Kenton Lee, Kristina Toutanova

2) Transformer paper "Ashish Vaswani, Noam Shazeer, Niki Parmar, Jakob
Uszkoreit, Llion Jones, Aidan N Gomez, Lukasz
Kaiser, and Illia Polosukhin. 2017. Attention is all
you need. In Advances in Neural Information Processing Systems, pages 6000–6010."

Experimental design

In the experiment, author compared their approach with various classical ML approaches which seems to be the right approach.

In these binary classification, usually there is a trade-off between precision and recall. I wonder why author did not share the precision and recall numbers separately instead of only sharing f1 score.

I would recommend sharing precision and recall numbers as well. This will further help the audience to understand the performance of various approaches.

Validity of the findings

Paper provides the details of various hyper-parameters used in the experiments.
Given that author extensively compared the novel approach with the existing classical ML methods, it shows the approach proposed has some advantages.

Raw data has not been shared but the aggregated metrics, details on hyper-parameters and various metric plots prove that results are valid.

Cite this review as

Reviewer 3 ·

Basic reporting

The work presented in this paper points to the prospect using modern natural language processing model to solve the problem of machine learning code classification. This paper presents significant work, however, authors should present their contribution in a better way in the Abstract section. In addition, there are several issues that are required to be addressed, including:
1. Discuss the main problem and your solution in the Introduction Section.
2. Add more relevant works to Section 2 (Literature Review).
3. The main concept of this work was not well presented. Authors need to add more details about the system design.
4. Authors must discuss the results presented in Figures 12, 13, and 14.

Experimental design

Authors compared the transformer-based and classical ML models to classify the ML code snippets. However, the obtained results need to be discussed. In addition, authors need to analyse the obtained results and compare them with the results obtained from the relevant previous approaches.

Validity of the findings

Additional discussions are required to be presented. In addition, I suggest adding a new Section named as Discussion that involves analyzing the obtained results.

Cite this review as

---

## Round 0.2 · accepted · Accept

As per comments from reviewers, this revised paper can be accepted.

Reviewer 3 ·

Basic reporting

Authors have addressed the raised issues.

Experimental design

Authors have addressed the raised issues.

Validity of the findings

Authors have addressed the raised issues.

Additional comments

Authors have addressed the raised issues.

Cite this review as

·

Basic reporting

The author of this technical paper has demonstrated exceptional proficiency in the English language. The paper exhibits a high degree of organization, making it accessible and detail oriented for a wide range of audiences. One minor correction or clarification needed in Line 131 and 132: “"The experiments and results are described in Section ." This line is missing section number, so that the readers would know what section to refer.

Experimental design

Author’s work sheds light on improvising the already explored territory of enriching ML specific code from annotation perspective. Paper provides valuable insights, data, and approaches that future research can learn for effectively using modern transformers codeBERT. Author’s work clearly exhibits the usage of codeBERT model performing better than classic ML models with relevant metrics. One minor comment for author would be, Beyond the Python-specific corpus, it would be beneficial to provide additional information for a broader audience regarding the application of this methodology to other programming languages. The author's insights into how this process can be adapted for different programming languages would greatly benefit a wider readership, extending the relevance of their work beyond the Python-specific context. The author's explanation of hyperparameter selection for various classification algorithms is cogent and concise. line numbers 237, 238, and 239 says how well the authors took care of the test set that's suitable for testing; it's impressive that they anticipated such circumstances and made an effort to evade the noisy test set.

Validity of the findings

The author has demonstrated meticulous consideration of corner cases and purposefully omitted them while considering the training and validation data to enhance the accuracy and robustness of the observed and presented results. Furthermore, a clear and concise comparison of metrics between the classic machine learning model and the CodeBERT model enables readers to readily grasp the tangible level of enrichment that CodeBERT can provide. This side-by-side evaluation serves as a valuable reference point, illustrating the significant advantages and enhancements brought about by the CodeBERT model in a manner that is easily comprehensible to the audience.

Additional comments

Improvements:
The authors would have experimented more with the current popular transformer models apart from just BERT. We have a huge list of BERT families for sequence classification like XLNet and RoBERTa, which is good for fewer classes such as this paper's case. The experiment results would have been more plausible if the achieved accuracy had been compared with more baselines
Detailed investigation and explanation on FP, and FN would've been more useful for future research, as it throws light on the present research gap and future improvement from this point.
In the preprocessing step, a supporting result-based claim for removing the redundant information would've made a strong assertion on the preprocessing claims; I strongly recommend a kind of ablation study for stated redundant information removal from lines number 102 to 106. However, the presented information is enough to prove the combination of redundant information removal yields better results.

Cite this review as